# Traditional Mexican Food: Phenolic Content and Public Health Relationship

**DOI:** 10.3390/foods12061233

**Published:** 2023-03-14

**Authors:** Julia María Alatorre-Cruz, Ricardo Carreño-López, Graciela Catalina Alatorre-Cruz, Leslie Janiret Paredes-Esquivel, Yair Olovaldo Santiago-Saenz, Adriana Nieva-Vázquez

**Affiliations:** 1Instituto de Ciencias, Benemérita Universidad Autónoma de Puebla, Puebla 72570, Mexico; 2Department of Pediatrics, University of Arkansas for Medical Sciences, Little Rock, AR 72205, USA; 3Universidad del Valle de Puebla, Puebla 72440, Mexico; 4Área Académica de Nutrición, Instituto de Ciencias de la Salud, Universidad Autónoma del Estado de Hidalgo, San Agustín Tlaxiaca 42160, Mexico; 5Facultad de Medicina, Benemérita Universidad Autónoma de Puebla-Complejo Regional Sur, Puebla 72420, Mexico

**Keywords:** phenolic compounds, antioxidants, oxidative stress, chronic degenerative diseases

## Abstract

Phenolic compounds have a positive effect on obesity, diabetes, and cardiovascular diseases because of their antioxidant and anti-inflammatory capacity. The prevalence of these diseases has increased in the last years in the Mexican population. Therefore, the Mexican diet must be assessed as provider of phenolic compounds. To assess this, a survey of phenolic compound intake was validated and applicated to 973 adults (798 females) between 18 and 79 years old. We compared the phenolic compound intake of 324 participants with more diseases (239 females) and 649 participants with healthier condition (559 females). The groups differed in sex, age, and scholarship. Males, older participants, and those with lower schooling reported suffering from more diseases. Regarding phenolic compound intake analyses, the participants with healthier conditions displayed a higher phenolic compound intake than the other group in all foods assessed. In addition, the regression model showed that the phenolic compounds intake of Mexican dishes, such as arroz con frijol or enchiladas, positively affected health status, suggesting that this traditional food is beneficial for the participant’s health condition. However, the weight effect of PCI was different for each disease. We conclude that, although PCI of Mexican food positively affects health conditions, this effect depends on sex, age, and participants’ diseases.

## 1. Introduction

In recent years, nutrition science has focused on counteracting nutrient deficiency and some diseases by identifying active-food components. Diet offers the possibility to improve the subject’s health conditions by using these components or functional food [1]. Functional food is part of a habitual diet, but it has special biological properties, such as phenolic compounds (PC). PCs are a diverse group of plant micronutrients [1], some of which modulate physiological and molecular pathways involved in energy metabolism [2]. They can act by different mechanisms, the most important of them are conducted by anti-inflammatory, antioxidant activities, and antiallergic [3]. The anti-inflammatory activity entails the activation of sirtuins; induction of Nrf2 via inhibition of microglial activation; and suppression of proinflammatory mediators (TNF-α, prostaglandins, C-reactive protein levels, interleukins IL-1α, IL-1β, IL6, IL8, and intercellular adhesion molecule-1), while the antioxidant activity captures unpaired electrons present in free radicals, interruption of autoxidation chain reactions, deactivation of singlet oxygen, mitigation of nitrosative stress, activation of antioxidant enzymes, or inhibition of oxidative enzymes [1]. PCs can be divided into flavonoid and non-flavonoid derivatives [2,4]. Flavonoid is the most important PC class and includes more than 5000 compounds [5] that seem to impact human health positively. Multiple authors have explained the underlying mechanism of PC effect on human health. Oxidative stress (OS) and inflammation triggered by increased OS are the cause of many chronic diseases in reactive-oxygen species (ROS) [6]. Although OS is part of normal cellular conditions (e.g., mitochondrial respiratory chain), this could damage other biological molecules [7]. Therefore, the human body must trigger antioxidant mechanisms to prevent cellular injury. Interestingly, flavonoids have a protective effect because they counteract OS by using at least four mechanisms; (1) reduction of free radical formation, (2) protection of α-tocopherol in Low Density Lipoprotein (LDL) from oxidation, (3) regeneration of oxidized α -tocopherol, and (4) chelation of metal ions such as Fe and Cu, preventing the consequent production of new free radical [1].

Moreover, recent studies explain how PC positively affect certain illnesses, such as obesity, diabetes, cardiovascular diseases, thrombocytopenia, or metabolic syndrome [1,8,9]. Several common features characterize these pathologies; among them, we can highlight the redox balance and a notable inflammatory response that strongly alters the biochemical and functional characteristics of the affected tissues.

Regarding obesity, it results from energy imbalance, increasing adipose tissue [10,11]. PC intake reduce the weight gain by reducing adipose tissue using lipolysis [1,10,11,12,13,14]. Others studies report that subjects with obesity show a low level of antioxidant enzymes (e.g., catalase, glutathione peroxidase, and glutathione reductase), which generates lower antioxidant capacity [11,13,15,16]. Therefore, PC intake might also increase their antioxidant capacity through free-radical scavenging activity. 

Significant evidence from epidemiological investigations showed that dietary PCs might manage and prevent type 2 diabetes (T2D) [1]. T2D disease consists of metabolism disturbances characterized by chronic hyperglycemia or impairment of insulin secretion or action [17]. Hyperglycemia increases ROS production by glucose auto-oxidation and nonenzymatic glycation processes, affecting the normal function of proteins and lipids. Therefore, high PC intake has been highly recommended for this disease management to counteract ROS [1]. In addition, PCs from coffee, guava tea, whortleberry, olive oil, propolis, chocolate, red wine, grape seed, and cocoa have been reported to show antidiabetic effects in T2D patients through increasing glucose metabolism and improving vascular function, as well as reducing insulin resistance and the HbA1c level [1].

On the other hand, cardiovascular diseases entail problems in the heart and blood vessels. Patients frequently show increased blood pressure, evidencing a disorder in the circulatory system [18]. PC intake improves endothelial dysfunction [19,20], decreases vascular OS [20,21], inhibits platelet aggregation and the oxidation of LDL, and reduces blood pressure, evidencing their counteracting capacity on cardiovascular disease symptoms [19,21]. 

Most interventions for these diseases have been focused on changing the patient’s diet, but they have failed to conform to eating habits associated with culture. It is well known that there are multiple dietary habits worldwide, many of which have been described as a good source of PC. For example, in the Western diet, fruit, vegetables, tea, wine, and cocoa products provide a mean intake of flavonoids of around 250 mg/d for the United States (US) adults [4,22]. Greek and Korean populations have similar flavonoid intake to the US population, with a mean between 250 and 320 mg/d. In contrast, the British population has a PC intake greater than other countries because they consume around 1000 mg/d [22,23,24]. However, there is no data about the PC intake of Latino American populations, even when these have outstanding gastronomic richness.

Traditional Mexican food is characterized by using grains, tubers, legumes, vegetables, and spices [25,26], most of which are rich in PC [27]. However, the Mexican diet has changed over the last decades because traditional food has been replaced with ultra-processed food with high-caloric values [28,29]. Moreover, some vegetables and fruits are preferably consumed after some kind of processing, affecting the quantity, quality, and bioavailability of the PCs [1]. Along with changes, diseases associated with eating habits have increased by more than 27% in the Mexican population [24,30]. Over the last two years, the National Health and Nutrition Survey reports that 72.2% of the Mexican population is overweight or obese [28,29]. In a posterior survey, they added that 30.2% of the adults have hypertension, and 15.6% suffer from diabetes. In addition, there is a high prevalence of dyslipidemia diseases, with 49% of the adult population suffering from high levels of triglycerides and 54.3% with high cholesterol levels [24,30]. Health reports have also correlated the increased prevalence of these diseases with a decreased intake of fruits and vegetables.

In this study, we expected to find a better health condition in participants with higher PC intake. Given that traditional Mexican food is not ultra-processed, and this seems to contain high levels of PC [25,26], we would expect that participants with a higher intake of beverages or Mexican dishes would show better health conditions than those with a lower intake.

## 2. Materials and Methods

### 2.1. Participants

Nine hundred and seventy-three adults (798 females; 175 males) between 18 and 79 years old were enrolled in this study. They were ethnic Mexican and native Spanish speakers and had at least nine years of education. In this study, health condition was obtained from volunteers through a survey; therefore, they were not explored by a physician. All participants were informed of their rights and provided written informed consent for participation in the study. This research was carried out ethically and was approved by the Benemérita Universidad Autónoma de Puebla. 

### 2.2. Procedure

This is a cross-sectional study with a non-probabilistic sample. We obtained the data from a self-administered food consumption survey, which was directed at the open public. In the survey, we requested the food intake frequency with a high level of phenolic compound [1].

#### 2.2.1. Survey of Food Consumption with a High Level of Phenolic Compounds

The survey assesses participants’ intake of food and health condition. This entails one hundred and twenty-four items distributed in nine sections: (1) identification data; (2) anthropometric data; (3) medical records; and (4) food intake frequency with a high level of PCs in the last month: fruits, vegetables, cereals, legumes, spices, beverages, and Mexican dishes (See Table A1). The survey was posted on social media (i.e., Facebook) or via WhatsApp or email.

#### 2.2.2. Validation

The survey was applied to the pilot group of 32 subjects, who reported a complete understanding of the items. They also reported being comfortable with all items and completion times. The pilot group responded to 100% of the questions. The statistical analysis for the survey’s validation was performed using a chi-square test. The factor analysis technique assessed the items with an orthogonal rotation “Varimax”. In this analysis, the factor weight of each item was at least 0.4 for all items. We also measured internal consistency of each item for each factor, exploring their reliability using Cronbach’s alpha (0.96).

### 2.3. Data Analysis Methods

#### 2.3.1. Clustering Analysis

A K-means clustering was performed to determine the participant’s health-condition level. The variables included were the body mass index (BMI); number of diseases (diabetes, hypertension, hypercholesterolemia, hypertriglyceridemia, kidney disease, and fatty liver); and number of gastrointestinal symptoms or illnesses: constipation, gastritis, irritable bowel syndrome (IBS), peptic ulcer, bacterial overgrowth syndrome (BOS), and ulcerative colitis (UC). The clustering analyses resulted in 649 participants with less diseases (LD) and 324 with more diseases (MD).

#### 2.3.2. Comparisons between LD and MD Groups

Demographic data. A chi-square test was used to compare groups for sex and age distribution. The groups were also compared for the scholarship, BMI, number of dietary supplements, number of diseases, and number of gastrointestinal symptoms or illnesses using independent *t*-tests. 

Phenolic compounds intake. We calculated the frequency of PC intake for each participant. For food intake frequency, we considered the number of times the food was consumed and its grams in the last month. We calculated the frequency of phenolic-compounds intake (PCI) using the food biochemistry composition reported in multiple papers (See Table 1), and selected only papers describing total phenolic compounds (TPC) mg gallic acid equivalent (GAE)/100 g. Given that TPC’s composition varies by multiple conditions, we computed the TPC’s average for each food, considering the variations (i.e., fruits, vegetables, cereals, legumes, seeds, spices, and beverages). Then, for each participant as follows: PCI = [(food-intake frequency × TPC)/participant’s BMI].

For Mexican dishes, we added the TPC of each recipe’s ingredient, then recalculated TPC for an individual portion of each recipe (TPCr). Given that some recipes of Mexican dishes are varied, we averaged the TPC of individual portions between recipes (See Table A2). For the analyses, we calculated phenolic compounds intake of recipe (PCIr) for each participant as follows:PCIr = [(food-intake frequency × Average of TPCr)/participant’s BMI].

ANOVAs were performed for each nutritional group (i.e., fruits, vegetables, cereals, seeds, spices, beverages, and Mexican dishes), and the sex or age category was considered as a between-subject factor, while PCI/PCIr was included as a within-subjects factor.

Two-way ANOVAs were performed for each nutritional group. Group (i.e., LD and MD) and the sex or age category was considered as a between-subject factor to observe the sex and age category effects, and PCI or PCIr was included as a within-subjects factor. Data were analyzed using SPSS Statistics 21. Greenhouse–Geisser corrections were made for violations of sphericity when the numerator was greater than 1, *p*-values resulting from a set of comparisons were corrected by the false discovery rate method (FDR). We report results surviving FDR correction (*p* values < 0.05).

#### 2.3.3. Risk of Developing Disease

Regression analyses were performed to identify the association between the participant’ diseases, PCI or PCr, and other variables. Linear regression was performed using as a dependent variable our cluster (i.e., LD and MD groups), and PCI, PCIr, sex, age, and scholarship were also included as independent variables. Linear regression was also performed per disease (i.e., diabetes, hypertension, hypercholesterolemia, hypertriglyceridemia, kidney disease, fatty liver, and obesity), including as the dependent variable the presence or absence of disease, and PCI, PCIr, sex, age, scholarship as independent variables. The linear regression analyses include multiple linear backward regressions to find a reduced model that best explains the data. A *p*-value < 0.05 was considered statistically significant in all analyses. 

## 3. Results

### 3.1. Demographic Results

Differences between groups were also observed in the sex (Xi (1) 22.4, *p* < 0.001) and age category distributions (Xi (4) 34.2, *p* < 0.001). The men’s distribution was greater than expected in the MD group (See Table 2). In contrast, in the age category, the groups differed in the subset of participants between 18 and 29 years old. A greater number of participants than the expected count was observed for the LD group, while the inverse pattern was observed for the MD group.

The groups were significantly different in scholarship (t (971) 3.7, *p* < 0.001). The LD group had greater years of schooling than the MD group (LD, Mean (M) = 15.4; MD, M = 16.0; Cohen’s d = 0.2). They also differ in BMI (t (971) −32.3, *p* < 0.001), with the LD group displaying lower BMI than the other group (LD, M = 22.4; MD, M = 30.1; Cohen’s d = −2.2). In addition, the groups did not differ in the number of dietary supplements consumed (t (971) 1.7, *p* = 0.1; LD, M = 0.8; MD, M = 0.7).

As we expected, the MD group also showed greater numbers of diseases than the LD group (t(971) −7.1, *p* < 0.001, Cohen’s d = −0.4; LD, M = 0.1; MD, M = 0.4). However, they did not differ in the number of gastrointestinal diseases or symptoms (t (971) 1.2, *p* = 0.2) (See Table 3).

### 3.2. Phenolic Compounds Intake Results

Regardless of the level of the health condition, participants had a greater intake of apples (67.2%), oranges (56.7%), tomatoes (86.7%), white onions (79.9%), chilies (71.5%), carrots (71.5%), lettuce (70.3%), nopal (58.2%), potatoes (57.7%), corn (74.2%), rice (67.8%), oatmeal (56.5%), and beans (65.9%). The beverages more frequently consumed were coffee (65.1%) and hibiscus water (54.6%), while the Mexican dishes more consumed were salsas verdes (61.9%), followed by salsas rojas (57.5%) (see Figure 1A). 

The sex groups differed in PCI of some foods. Females had a higher PCI of cereals, legumes, seeds, and beverages than males, while males showed a higher PCI of fruits. The age groups also differed in PCI. The participants between 40 and 49 years of age had a higher PCI of vegetables, those between 18 and 29 years old showed higher PCI of legumes, while the participants between 50 and 59 years of age had a higher PCI of spices than the other groups (see Table 4).

#### 3.2.1. Phenolic Compounds Intake: Group and Sex Distribution

Fruits. The groups were different in PCI of fruits, and a main effect of the group was found (F (1, 960) = 82.6, *p* < 0.001, η2p = 0.08, ε = 0.07); this evidenced a higher PCI of fruits for LD than MD groups (Mean difference, (Md) = 30.1, *p* < 0.001; LD, M = 139.3, MD, M = 109.2). A significant group by fruit interaction was also observed (F (15, 960) = 51.0, *p* < 0.001, η2p = 0.05, ε = 0.07). The post-hoc tests confirmed that the LD group had a higher PCI of each fruit than the MD group (Each fruit, *p* < 0.01). No significant group by fruits by sex interaction was found (See Figure 1B).

Vegetables. A main effect of the group was found (F (1, 960) = 77.6, *p* < 0.001, η2p = 0.07, ε = 0.07). The LD group showed higher PCI of vegetables than the MD group (Md = 9.2, *p* < 0.001; LD, M = 45.62, MD, M = 36.45). A significant group by vegetable interaction was also found (F (17, 960) = 43.9, *p* < 0.001, η2p = 0.04, ε = 0.06). The post-hoc tests revealed that the LD group had a higher PCI of each vegetable than the MD group (each vegetable, *p* < 0.001). A significant group by vegetables by sex was also observed (F (17, 960) = 4.46, *p* = 0.03, η2p = 0.005, ε = 0.06). However, the post-hoc test showed that female or male groups did not differ in PCI of vegetables between LD and MD groups (See Figure 1B).

Cereals. As is shown in Figure 1B, a main effect of group in PCI of cereals (F (1, 960) = 107.5, *p* < 0.001, η2p = 0.1, ε = 0.4) was observed. As we hypothesized the LD group had a higher PCI of cereals than the MD group (Md = 7.25, *p* < 0.001; LD, M = 35.69, MD, M = 28.44). A significant group by cereal interaction was observed (F (6, 960) = 59.22, *p* < 0.001, η2p = 0.06, ε = 0.4). The post-hoc tests confirmed a higher PCI for each cereal for LD than the MD group (Each cereal, *p* < 0.001). We also found a significant group by cereal by sex interaction (F (6, 960) = 5.68, *p* < 0.001, η2p = 0.006, ε = 0.4). Although, the pair comparisons showed that PCIs of cereals differed between LD and MD groups, with a greater PCIs for women that belonged to the LD group. The men’s subgroups, the LD and MD groups, did not show differences in PCIs of flaxseed, wheat, and millet (Flaxseed, Md = 4.8, *p* = 0.1; wheat, Md = 0.8, *p* = 0.2; millet, Md = 0.2, *p* = 1.0). 

Legumes. The groups were also different in the PCI of this nutritional group. A significant main effect of group (F (1, 960) = 150.4, *p* < 0.001, η2p = 0.1, ε = 0.6) evinced that LD group showed a higher PCI of legumes than the MD group (Md = 62.0, *p* < 0.001; LD, M = 268.6, MD, M = 206.5) (See Figure 1B). We also found a significant group by legume interaction (F (3, 960) = 88.3, *p* < 0.001, η2p = 0.08, ε = 0.6). The LD group showed a higher PCI of all legumes than the other group (Each legume, *p* < 0.001). In addition, a significant group by legume by sex was also observed (F (3, 960) = 3.6, *p* = 0.03, η2p = 0.004, ε = 0.6). Although the comparisons favored the LD group for the female subgroup, only for the male subgroup, the PCI of soybeans was not different between groups with a different health condition (Soybean, Md = 2.2, *p* = 0.1).

Seeds. A main effect of the group was also found in this nutritional group (F (1, 960) = 104.8, *p* < 0.001, η2p = 1.0, ε = 0.3). The LD group showed a higher PCI of seeds than the other group (Md = 38.6, *p* < 0.001; LD, M = 180.7, MD, M = 142.1) (See Figure 1B). A significant group by seed was also observed (F (4, 960) = 70.6, *p* < 0.001, η2p = 0.07, ε = 0.3). The post-hoc tests evidenced that the LD group displayed higher PCI of seeds than the MD group (Each seed, *p* < 0.001). No significant group by seed by sex interaction was found.

Spices. The groups were different in the PCI of spices (F (1, 960) = 62.3, *p* < 0.001, η2p = 0.06, ε = 0.1). The LD group had a higher PCI of spices than the MD group (Md = 12.3, *p* < 0.001; LD, M = 67.8, MD, M = 55.6) (See Figure 1B). A significant group by spice interaction confirmed that the LD group had a higher PCI than the other group in each spice [F (19, 960) = 14.5, *p* < 0.001, η2p = 0.01, ε = 0.1]. In addition, a significant group by spices by sex interaction was also observed (F (19, 960) = 5.0, *p* = 0.002, η2p = 0.05, ε = 0.1). Only for male group multiple PCI of spices were no different between the LD and MD groups (Marjoram, Md = 13.0, *p* = 0.5; achiote (annatto), Md = 0.6, *p* = 0.1; Chaya, Md = 0.1, *p* = 1.0; fennel, Md = −0.2, *p* = 0.7; linden, Md = 2.0, *p* = 0.8; saffron, Md = 1.7, *p* = 0.7; Mexican pepper leaf, Md = 4.9, *p* = 0.06; papalo, Md = 6.2, *p* = 0.2). The remaining comparisons were significant in favor of the LD group regardless of sex. 

Beverages. The groups with different health condition differed in PCI of beverages (F (1, 960) = 111.46, *p* < 0.001, η2p = 0.3, ε = 0.1). The LD groups showed a higher PCI of beverages than the other group (Md = 145.24, *p* < 0.001; LD, M = 759.26, MD, M = 614.01) (See Figure 1B). A significant group by beverages was also found (F (4, 960) = 45.94, *p* < 0.001, η2p = 0.3, ε = 0.5). The post-hoc tests evidenced that the LD group had a greater PCI in all beverages than the MD group (*p* < 0.01). No significant group by beverage by sex was observed.

Mexican dishes. A main effect of the group was observed (F (1, 960) = 196.6, *p* < 0.001, η2p = 0.2, ε = 0.4). The LD group displayed a greater PCI of Mexican dishes than the other group (Md = 47.7, *p* < 0.001; LD, M = 204.6, MD, M = 156.9) (See Figure 1B). A significant group by Mexican dishes interaction (F (12, 960) = 47.6, *p* < 0.001, η2p = 0.04, ε = 0.4) confirmed that the LD group had a higher PCI of each Mexican dish (*p* < 0.001). No significant group by beverage by sex was observed.

#### 3.2.2. Phenolic Compounds Intake: Group and Age Distribution

Although, no significant group by age category interaction was observed for any comparison, but a significant main effect of group (i.e., LD vs. MD) was observed for each comparison with LD showed higher PCI than MD group: fruits (F (4, 954) = 55.02, *p* < 0.001, η2p = 0.05, ε = 0.07 (Md = 28,57, *p* < 0.001; LD, M = 134.43, MD, M = 105.85)), vegetables (F (4, 954) = 108.44, *p* < 0.001, η2p = 0.10, ε = 0.06 (Md = 12.26, *p* < 0.001; LD, M = 48.90, MD, M = 36.64)), cereals (F (4, 954) = 87.35, *p* < 0.001, η2p = 0.08, ε = 0.41 (Md = 7.56, *p* < 0.001; LD, M = 34.97, MD, M = 27.41)), legumes (F (4, 954) = 129.57, *p* < 0.001, η2p = 0.12, ε = 0.56 (Md = 66.51, *p* < 0.001; LD, M = 270.57, MD, M = 204.06)), seeds (F (4, 954) = 104.59, *p* < 0.001, η2p = 0.10, ε = 0.33 (Md = 44.40, *p* < 0.001; LD, M = 186.51, MD, M = 142.11)), spices (F (4, 954) = 16.58, *p* < 0.001, η2p = 0.07, ε = 0.15 (Md = 15.21, *p* < 0.001; LD, M = 71.43, MD, M = 56.22)), beverages (F (4, 954) = 97.46, *p* < 0.001, η2p = 0.3, ε = 0.9 (Md = 156.30, *p* < 0.001; LD, M = 767.05, MD, M = 610.75)), and Mexican dishes spices (F (4, 954) = 156.49, *p* < 0.001, η2p = 0.14, ε = 0.37 (Md = 49.15, *p* < 0.001; LD, M = 204.92, MD, M = 155.76)).

### 3.3. Risk of Developing Diseases

As shown in Table 5, two regression models displayed an R^2^ higher than 0.4, the remaining regressions had a R^2^ of 0.1 (See Table A3). The stronger regressions included as independent variables our cluster (i.e., LD and MD groups) and obesity disease. In the regression model, which included our cluster, we found that high PCI of tomato, garlic, lettuce, corn, grape, wine, romeritos, arroz con frijoles, and scholarship predicted a smaller number of diseases. In contrast, older age and higher PCI of wheat and cranberry predicted a higher likelihood of suffering from a disease (See Figure 2). In the regression model, which included obesity as an independent variable, PCI of tomato, corn, garlic, chamomile tea, coffee, grape, Swiss chard, enchiladas, and wine predicted the absence of obesity, while the PCI of plum and oregano predicted the presence of this disease (For further information see Table A4 and Table A5).

## 4. Discussion

This study examined the PCI and how the Mexican diet was associated with participants’ health condition. We expected to find more diseases in participants with lower PCI based on previous literature. Given that traditional Mexican food is not ultra-processed and contains high levels of PC [25,26], we expected better health condition in participants with a higher intake of beverages or Mexican dishes. Our hypotheses partially agreed with our results because food with high PC was associated with better health condition; however, the consumption of beverages and Mexican dishes is lower than our expectations. 

### 4.1. Demographic Data

Our statistical analyses revealed that sex, age, and scholarship variables seem to play a role in the presence or absence of diseases in our Mexican cohort. Males suffer more diseases than females even when we had more participation from females in our sample However, this result matched previous studies reporting that males have lower life expectancy than females in Mexico [123]. A recent study explained that males do not usually go to the doctor when they have symptoms of illness, making them a vulnerable population to develop multiple diseases. As the prior literature has reported, our participants over 29 years had more diseases than their younger pairs. This result confirmed that older age increases the likelihood of suffering a disease due to the natural deterioration of the human body [124]. The scholarship also seems to influence disease development; those participants with higher scholarship reporting less illnesses. Jun et al. (2016) described that participants with higher scholarship usually include dietary supplements. In our study, even when we did not explore the relationship between scholarship and number of dietary supplements, we suggest that our participants with high scholarships could have nutritional habits such as Jun et al.’s participants. In addition, our demographic interpretations should be carried out carefully because our sample is not representative of the Mexican population.

### 4.2. Frequency of Food Consumption

Mexican eating habits of fruits matched the US and Korean populations in apple and orange consumption [24,30], and it was also consistent with the eating habits of Asian populations, mainly in chilies and rice consumption [24], our sample’s data also matched European Union in wheat and potato intake [125]. We suggest that Mexican’s eating habits match other populations because of the increased influence of other countries via social media and economic globalization. 

Previous studies reported that the Mexican population has lower consumption of without or low sugar beverages and traditional Mexican dishes than those described in the last decades [28,29], suggesting higher consumption of fast or ultra-preprocessed food. In our study, more than 50% of our sample consumed only two traditional Mexican dishes (i.e., salsas verdes and salsas rojas), and the most consumed beverage was not autochthonous from Mexico (e.g., coffee). Moreover, their consumption was similar to the eating habits of Latin American and Asian populations [1]. We could suggest that our data confirm the alarming changes previously reported in the Mexican diet. However, we also found that beans, corn (mainly tortilla), and nopal intake remained preserved in the Mexican eating habits, with a higher intake than in other countries [1]. These foods are essential in other autochthonous Mexican dishes such as “tacos”. Therefore, we would suggest exploring other Mexican dishes in future studies.

### 4.3. Comparisons between LD and MD Groups

The LD and MD differed in the PCI of all food groups, with LD showing a higher PCI than MD groups. These results matched our hypothesis and previous studies reporting a positive effect of PC on different diseases [22,23,24]. Vegetables and fruits have been considered great providers of PC [1]. Then, we expected a higher intake for LD group. The vegetables consumed more frequently by our sample, such as tomato and lettuce, were good predictors of a smaller number of diseases, which matched our hypothesis. However, even when the PCI of fruits was higher for the LD group, we found that low percentage of our sample consumed fruits. Moreover, the fruits consumed by more than 50% of our sample were not variables predictors in the regression analyses (See Figure 2). The lower PCI of fruits observed in this sample might be part of the problem in the health status of the Mexican population; we suggest that they might be replacing fruit with high-caloric snacks. Given that, in this study, we did not explore that kind of food, this statement would require further research.

We also found that cereals were consumed more by LD than MD groups. Corn is the most consumed by our sample; this cereal contains PC, which seems to decrease the risk of developing a chronic illness, such as diabetes, and cardiovascular diseases. Corn contains a molecule called lignin, which is the main component of dietary fiber. In addition, it has two functions: (1) inhibits enzymatic activity associated with the production of radical anion superoxide and (2) blocks the growth and viability of cancerogenic cells [126]. Then, cereals, particularly corn consumption, seems to affect health status positively. This statement is consistent with the regression analysis results, in which high corn intake was associated with a lower number of diseases in our sample.

Our statistical analyses also showed that LD had a higher PCI of legumes than MD groups. Previous study report that the legumes have a high biologic value because they contain essential amino acids and PC. These can modify the basal physiological function within the intestinal microenvironment affecting the microbiota and epithelial barrier, improving metabolic and gastrointestinal health, enhancing resistance to colonization by pathogens, and exerting an impact on the gut microbiota. They regulate metabolic stability and membrane transport in the intestine, thus improving bioavailability. These actions decrease the severity of diseases associated with the intestine due to their chemopreventive effects [127]. In our study, the analyses revealed no differences between LD and MD groups in gastrointestinal symptoms and diseases. We suggest that the positive effect of legumes on the digestive system might have been hampered by the poor variety of legumes consumed by our sample. They mainly consumed beans. On the other hand, legumes also contribute to glycemic control and protein anabolism [127]. LD and MD differed in the presence of diseases, between them diabetes mellitus. Interestingly, the Mexican dish “Arroz con frijoles” contains as a main ingredient bean, and its high intake was associated with fewer diseases. A recent study reported that beans improve postprandial glycemic response and glycated hemoglobin (HbA1c) by inhibiting α-amylase, and maltase. Therefore, beans have been considered an ideal food for managing blood glucose and insulin resistance [128]. Our results suggest that a high bean intake positively affects glycemic control in our sample.

We also found that LD and MD groups differed in the PCI of seeds, with a higher PCI for LD group. The seeds are food groups that contain PC and modulator molecules, such as essential fatty acids, which protect the digestive tract and allow appropriate maintaining in lipid metabolism (i.e., a decrease of triglycerides, LDL, cholesterol, very low density lipoprotein (VLDL), regulation of markers of platelets such as a reduction of endothelial adhesion, platelet aggregation, decrease of inflammation markers related to arachidonic acid, modulating the production of prostaglandins, leukotrienes, decrease of cyclooxygenases, reduction of oxidation of molecules such as LDL, deoxyribonucleic acid (DNA), reduce the production of ROS, increase reduced glutathione (GSH), glutathione peroxidase (GSH-Px), and plasma antioxidant capacity) [129]. In our study, the seeds were not consumed by more than 50% of our sample, which might be explained by their high cost and scarce availability in the market. This fact might justify that PCI of any seed was a predictor variable in our regression model, suggesting that this food did not reach a significant effect on health status because of its low consumption. A similar situation was found in the PCI of spices, even when they have many properties such as digestive stimulant action, antimicrobial, anti-inflammatory, antimutagenic, anticarcinogenic potential, and antioxidant capacities [130]. They were not consumed by more than 50% of our sample; therefore, they did not reach a place between predictor variables. We suggest that most of the participants in our study did not know whether their meals had spices. The participants’ report might be biased by their lack of knowledge of the dish’s ingredients.

As we expected the PCI of beverages were higher for LD than MD groups. Particularly, coffee was the beverage more consumed by our participants. This beverage contains chlorogenic acids (CGA), which have several effects on health conditions; between them, they are hypoglycemic, antiviral, and hepatoprotective and have antispasmodic activities [131]. In addition, a daily coffee intake of 2.5 cups has beneficial effects on endothelial function and vascular smooth muscle function in patients with hypertension [1], while another study reported that the level of coffee intake was not associated with gastrointestinal diseases and gastric cancer [1]. Moreover, in prior meta-analyses, the effect of coffee intake on obesity and chronic diseases is still controversial. However, a positive association between coffee intake, BMI, and abdominal obesity has been reported, suggesting several biological mechanisms against obesity triggered by biologically active compounds in coffee, such as CGA, caffeine, trigonelline, and magnesium [1]. In a study using an animal model, they reported that supplementation with CGA was associated with a body-weight reduction, a decrease of visceral fat mass, and lower triglyceride content in adipose tissue in high-fat-fed mice [1].In an in vitro study, trigonelline inhibited adipocyte proliferation and lipid accumulation in differentiating adipocytes [132]. In our study, the PCI of coffee was a predictor variable, that is, high PCI of coffee was associated with less number of diseases and the absence of suffering obesity disease, which matched previously described [1].

Although wine consumption was low in our sample, we found a relationship between fewer diseases or healthier weight status and wine intake. This result might be supported by recent studies describing the beneficial effects of wine compounds on health status. Verajano and Lujan-Corro (2022) explain that red wine contains at least two kinds of PC: flavonoids (anthocyanins, flavanols, and flavonols) and non-flavonoids (stilbenes, phenolic acids, among others); these PCs are attributed with antioxidant activity by increasing the activity of the catalase, superoxide dismutase (SOD), and glutathione reductase (GR) enzymes, or by enhancing the production of nitric oxide (NO), with a consequent lower cardiovascular risk. In particular, wine quercetin and resveratrol compounds can bind to LDL through glycosidic bonds lowering their levels and increasing levels of high-density lipoprotein protecting them against free radicals and reducing oxidation induced by metal ions [1]. In addition, these compounds together with tannic acid, and malvidin may also improve endothelial NO production, reducing platelet aggregation and vascular inflammation, while anthocyanins seem to reduce LDL levels in patients with dyslipidemia [1]. In this study, our findings suggest that wine prevents multiple mechanisms associated with the occurrence of diseases, which might explain our results [1].

In this study, chamomile tea showed a positive effect on the health status. Prior studies have described that chamomile flower head has several flavonoids such as apigenin, quercetin, patuletin, and luteolin. Moreover, the chamomile hot water extract and its major components (esculetin and quercetin) show moderate sucrase inhibition, suggesting that this influences the prevention of hyperglycemia in diabetics patients [133]. In addition, an animal model study reports that chamomile has a potent anti-inflammatory action, antimutagenic and cholesterol-lowering activities, and antispasmodic and anxiolytic effects [1]. Given that chamomile was a predictor variable for a smaller number of diseases. These properties attributed to chamomile might support the results observed in this study.

We expected that PCI of Mexican dishes would be higher for LD than MD participants, and our data matched our hypothesis. However, the most consumed dishes (i.e., salsas rojas and salsas verdes) were not predictor variables in our regression model. Instead, Arroz con frijoles was associated with a smaller number of diseases. As we mentioned, the beans are part of the Arroz con frijoles’ recipe. We suppose that the bean cooked in the traditional way might promote better health conditions in our Mexican sample [128].

PCI of enchiladas was also a predictor variable, which was associated with the absence of suffering from obesity. We suggest that this dish has beneficial properties on health conditions because its main ingredients are tortillas of corn and tomato. Previous studies have described that corn mainly entails ferulic acid, followed by p-coumaric acid, which are highly copious in their conjugated forms [1]. In an animal model was verified that dietary ferulic acid supplementation suppresses blood glucose elevation, body and hepatic lipid accumulation, body weight gain, and inflammatory cytokines (IL-6 and TNF-α) in high-fat diet-induced obese mice, suggesting that ferulic acid could be helpful in lowering the risk of high fat-diet induced obesity and obesity-related metabolic syndromes [1]. In Mexico, white maize is the most consumed, and this is processed as tortillas. This food implies nixtamalization process, which entails the hydrolysis of the ester linkage between the ferulic acid and the cell wall components, which triggers the soluble fraction to be higher in nixtamalized maize products relative to that found in the whole grain—that is, ferulic acid increases 26% [1]. Therefore, tortillas intake might have a positive effect on health condition of Mexican population. On the other hand, a recent study reports that tomato has multiple properties; between them, this improves the antioxidant defense and plasma lipid peroxidation products, the lipid profile, and HbA1c levels [134,135]. Therefore, both ingredients help to prevent the development of abnormal weight status and other diseases. 

## 5. Conclusions

We conclude that PCI positively affects health conditions and supports the hypothesis that specific nutritional foods have a particular effect on certain diseases. For example, higher PCI of arroz con frijoles was associated with lower number of diseases. In contrast, a higher PCI of enchiladas was associated with a lower likelihood of suffering from diabetes. Therefore, the specific effects of Traditional Mexican food will need further research.

As expected, the most consumed foods in our cohort positively affected the health conditions (e.g., tomatoes, lettuce, or corn), suggesting that the feeding habits of our Mexican sample promote a greater health status. On the other hand, it is essential to highlight that the foods less consumed had a solid effect to consider them as predictor variables (e.g., garlic, grapes, Swiss chard, or wine). Therefore, these foods must have specific properties which should be studied deeply. In addition, our findings suggest that the PCI effect also depends on sex, age, and local feeding habits. Therefore, a more controlled study should be performed to describe these effects precisely.

## 6. Limitations

There are inherent limitations in the present study. This is cross-sectional study, which may not support interpretations of causality between PCI and diseases. In addition, given that a physician did not explore our participants because the survey was applied during the pandemic virus (COVID-19). The participants might suffer from other health conditions. Therefore, interpretations should be carried out carefully. In addition, we did not use the traditional daily intake survey, anthropometric measures, or biochemical indices to assess the participants; these might provide more nuanced metrics for studies examining the impact of PCI on diseases. The PCI is not a precise measurement, because our calculus was not performed considering the culinary technique for each Mexican dish. This fact may affect the amount of PC in each personal portion. In addition, we did not assess all Mexican dishes. Therefore, our conclusions are limited to the more common dishes.

## Figures and Tables

**Figure 1 foods-12-01233-f001:**
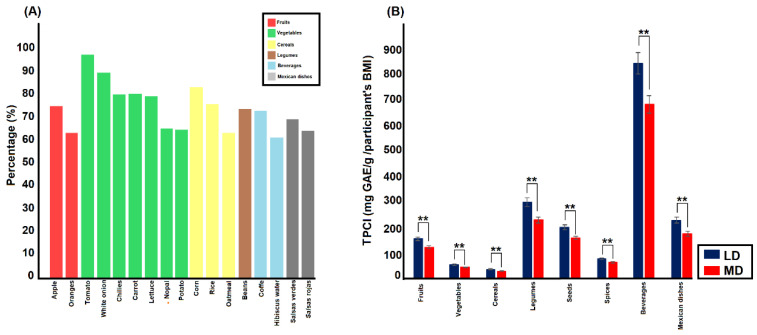
Bar graph (**A**) shows food more frequently consumed in this Mexican sample, while bar graph (**B**) illustrates differences in the total phenolic compounds intake (PCI) of all food groups between LD and MD groups. ** *p* < 0.001.

**Figure 2 foods-12-01233-f002:**
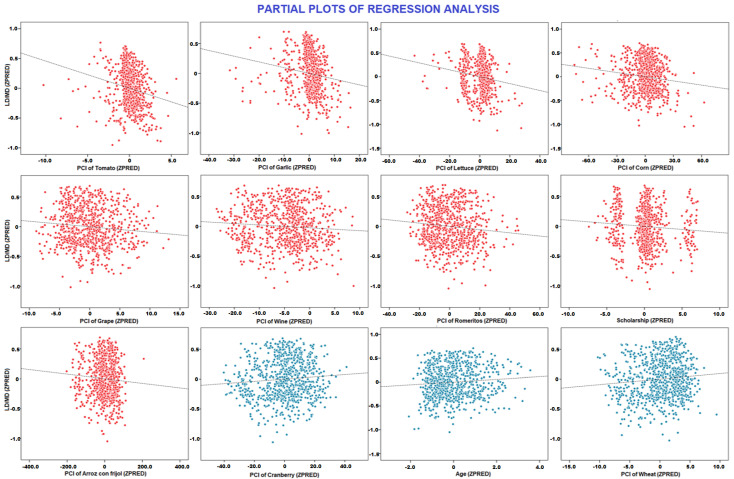
The scatter plots illustrate the relationship between LD and MD groups and predictor variables in the regression model. Negative associations are in red, and the positive ones are in blue.

**Table 1 foods-12-01233-t001:** Total phenolic compounds (TPC) of each food.

	Scientific Name	TPC mg GAE/100g (A)	References
Fruits			
*Grape*	*Vitis vinifera* L. *“Red Globe”*	122.2	[31]
*Plum*	*Prunus domestica*	219.9	[32]
*Cranberry*	*Vaccinium subg. Oxycoccus*	392.37	[33]
*Peach*	*Prunus persica* L.	74.5	[34]
Raspberry	*Rubus idaeus*	248.25	[35]
Blueberry	*Vaccinium sect. Cyanococcus*	335	[33]
Prickly pear	*Opuntia ficus indica* L.	94.3	[36,37,38,39]
Apple	*Malus domestica*	131.95	[40]
Pink grapefruit	*Citrus paradisi* L.	192.0	[41]
Kiwi	*Actinidia deliciosa*	47.0	[42]
Orange	*Citrus sinensis* L.	2325.0	[43]
Guava	*Psidium guajava*	318.5	[27]
Strawberry	*Fragaria X ananassa*	389.6	[44]
Pomegranate	*Punica granatum* L.	15,699.0	[45]
Cherry	*Prunus avium* L.	124.0	[46]
Pear	*Pyrus communis* L.	49.68	[47]
Vegetables			
Huitlacoche (Fungi)	*Ustilago maydis*	53.0	[48]
Spring onion	*Allium fistolisum*	72.7	[49]
Mushroom (Fungi)	*Agaricus bisporus*	287.4	[50]
Peppers	*Capsicum annuum* L.	160.2	[27]
Carrot	*Daucus carota* L.	70.1	[51]
Beetroot	*Beta vulgaris* L.	207.5	[52]
Swiss chard	*Beta vulgaris* L.	126.0	[53]
Tomato	*Solanum lycopersicum* L.	76.5	[27]
Chilies	*Capsicum frutescens*	286.7	[27]
Lettuce	*Lactuca sativa* L.	3.13	[54]
Celery	*Apium graveolens* L.	4640.0	[55]
Brussels sprouts	*Brassica oleracea var. gemmifera*	192.0	[56]
Nopal (Prickly pear cactus)	*Opuntia Streptacantha and Fuliginosa*	17.13	[57]
Red Radish	*Raphanus sativus* L.	68.0	[58]
Broccoli	*Brassica oleracea* L.	106.0	[58]
White onion	*Allium cepa blanc* L.	24.3	[59]
Purple onion Potato	*Allium cepa* L. *Solanum tuberosum* L.	42.7 13.8	[59] [60]
Cereals			
Rice cooked	*Oryza sativa* L.	23.8	[61]
Oatmeal	*Avena sativa* L.	144.0	[62]
Barley	*Hordeum distichon* L.	138.5	[63]
Flaxseed	*Linum usitatissimum* L.	469.0	[64]
Wheat	*Triticum aestivum*	87.25	[63]
Corn	*Zea mays* L.	672.54	[59]
Millet	*Pennisetum glaucum*	6000.0	[65]
Sorghum	*Andropogon sorghum* L.	609.52	[66]
Legumes			
Soybean	*Glycine max* L.	212.04	[67]
Beans	*Phaseolus vulgaris* L.	3421.2	[68]
Haba	*Vicia faba* L.	106.3	[69]
Lentil	*Lens culinaris* L.	3846.5	[70]
Seeds			
Cocoa powder	*Theobroma cacao* L.	1104.5	[71]
Almont	*Prunus dulcis*	3795.5	[72]
Nut	*Juglans regia* L.	1383.5	[73]
Peanut	*Arachis hypogaea* L.	379.0	[73]
Chia	*Salvia hispanica* L.	116.0	[74]
Flaxseed	*Linum usitatissimum* L.	2310.0	[75]
Spices			
Parsley	*Pretroselinum crispum*	215.0	[76]
Coriander	*Coriandrum sativum* L.	138.0	[77]
Oregano	*Lippia graveolens*	441.0	[78]
Epazote	*Chenopodium ambrosioides* L.	1198.6	[79]
Garlic	*Allium sativum* L.	240.5	[80]
Clove	*Syzygium aromaticum*	896.0	[81]
Paprika	*Capsicum annuum* L.	368.5	[71]
Marjoram	*Origanum majorana* L.	2770.0	[82]
Eucalyptus	*Eucalyptus camaldulensis*	1412.0	[83]
Achiote (Annatto)	*Bixa Orellana* L.	73.0	[84]
Sesame Seed	*Sesamun indicum* L.	10.67	[85]
Ginger	*Zingiber officinale Roscoe*	1280.5	[86]
Black pepper	*Piper nigrum*	4.85	[87]
Chaya	*Cnidoscolus aconitifolius*	634.0	[88]
Fennel	*Foeniculum vulgare Miller*	123.7	[89]
Linden	*Tilia cordata*	1118.4	[90]
Saffron	*Crocus sativus*	610.0	[26]
Anise	*Pimpinella anisum*	298.6	[81]
Mexican pepper leaf	*Piper auritum Kunth*	398.1	[91]
Papalo	*Porophyllum ruderale*	680.4	[91]
Beverages			
Coffee	*Coffea*	18,500.0	[92]
Green tea	*Camellia sinensis* L.	218.1	[93]
		8916.6	[82]
Wine	*Vinum*	260.0	[94,95]
Hibiscus water	*Hibiscus sabdariffa* L.	3742.0	[96]
Chamomile tea	*Matricaria chamomilla* L.	3002.8	[97]
Mexican dishes		TPCr	
Mole ^1^	*Mole rojo* *Mole verde* *Mole Poblano* *Mole de olla*	4834.9 561.5 4116.3 5667.5 A: 3795.1	[27,59,71,72,73,76,79,80,81,85,87,91,98,99,100,101,102,103,104,105,106]
Arroz con frijol ^2^		1591.0	[51,59,61,68,79]
Enfrijoladas ^3^		1619.0	[27,59,68,79,98,105,107]
Enchiladas ^4^	*Enchiladas rojas*, *Enchiladas verdes*, *Enchiladas de olla*, *Enchiladas suizas*, *Enchiladas de verdolagas con requesón*	11,111.0 865.4 1603.7 620.0 3822.2 A: 3604.5	[27,54,59,71,77,78,80,87,98,105,107,108,109,110,111]
Salsas rojas ^5^	*Salsa Taquera*, *Salsa de chile habanero con tomate*, *Salsa Ranchera*, *Salsa de chile morita*, *Salsa de chile piquín.*	618.9 541.6 108.4 716.4 4517.7 A: 1300.6	[27,59,71,78,80,82,87,109,112,113]
Salsas verdes ^6^		1504.8	[27,59,80]
Cochinita pibil ^7^		1177.1	[43,78,80,81,84,101,114,115,116]
Huazontles ^8^		880.0	[27,59,71,80,117]
Quintoniles ^9^		912.1	[27,59,80,98,117]
Pipián ^10^		345.6	[27,71,72,81,85,87,91,98,99,104,118,119]
Romeritos ^11^		369.0	[27,57,59,60,71,73,80,81,85,98,99,102,104,105,117]
Verdolagas ^12^	*Verdolagas con espinazo*, *Verdolagas en salsa*, *Verdolagas en ensalada.*	6156.7 1454.7 1247.3 A: 2952.9	[27,51,55,59,76,80,81,87,91,101,108,109,111,112,115,120]
Ensalada con ^13^ espinacas		1339.3	[71,73,87,110,121,122]

TPC = Total phenolic compounds; GAE = gallic acid equivalent; A = average; TPCr = total phenolic compounds for an individual portion of each recipe. ^1^ Mole sauce, a sauce unique to Mexico, is often used in traditional Mexican dishes. Cocoa and spices come to the fore in this sauce, which is especially suitable for chicken meat. The ingredients are mixed until a smooth consistency is obtained. It can be stored by filling in can jars or bottles, if desired. ^2^ Arroz con frijoles is an essential dish of the Mexican diet. It entails a mix of beans and rice previously boiled; it is seasoned with salt and spices. ^3^ Enfrijoladas is a Mexican dish made using two essential ingredients: tortilla and beans. The tortillas are submerged in a sauce of mashed beans and filled with fresh cheese. It usually has a topping of vegetables (lettuce, tomato, or onion). ^4^ Enchiladas is a Mexican dish made using tortilla and Mexican sauce. The tortillas are submerged in a tomatoes and spice peppers sauce, and then, they are filled with chicken or vegetables. It usually has a topping of vegetables (lettuce, tomato, or onion). ^5^ The salsa roja is a Mexican dressing made with tomatoes, spicy peppers, and spices. ^6^ The salsa verde is a Mexican dressing made with tomatillo, spicy peppers, and spices. ^7^ Cochinita pibil is a Mexican dish prepared with pork meat marinated in achiote and wrapped in banana leaves; traditionally, it is cooked underground. ^8^ Huazontles is a Mexican stew with huazontles wrapped with egg and cheese. ^9^ Quintoniles is a Mexican casserole made with quintoniles sauteed with sauce and Mole Poblano. ^10^ Pipian is a Mexican dish with toasted pumpkin seeds, spices, pork, or turkey mead. ^11^ Romeritos is a Mexican dish made with romeritos leaves, shrimp, and Mole Poblano. ^12^ Verdolagas is a Mexican stew made with purslane, and pork mead, covered with a green sauce. ^13^ Ensalada de espinacas is a Mexican salad with spinach, oil seeds, spices, olive oil, and sour cream.

**Table 2 foods-12-01233-t002:** Sex and age range of participants.

Sex	Group	n (F/M)	% (F/M)
	LD	559/90	86.1/13.9
	MD	239/85	73.7/26.2
Age category		n	%
18–29	LD	333	51.3
	MD	107	33.0
30–39	LD	168	25.9
	MD	95	29.3
40–49	LD	86	13.3
	MD	70	21.6
50–59	LD	44	6.8
	MD	38	11.7
>59	LD	18	2.8
	MD	14	4.3

F = female, M = Male, n = number of participants, % = percentage, LD = Group with less diseases, and MD = Group with more diseases.

**Table 3 foods-12-01233-t003:** Distribution of diseases in the LD and MD groups.

	Group	
	LD	MD
Diseases	n (%)	
Diabetes mellitus	17 (2.6)	14 (4.3)
Hypertension	14 (2.2)	22 (6.8)
Hypercholesterolemia	13 (2.0)	31 (9.6)
Hypertriglyceridemia	19 (2.9)	34 (10.5)
Kidney disease	11 (1.7)	9 (2.8)
Fatty liver	8 (1.2)	28 (8.6)
Obesity	126 (19.4)	304 (93.8)
Symptoms of GD		
Constipation	138 (21.3)	82 (25.3)
Gastritis	152 (23.4)	90 (30.2)
IBS	115 (17.7)	45 (13.9)
Peptic ulcer	3 (0.5)	1 (0.3)
BOS	3 (0.5)	0 (0)
UC	2 (0.3)	0 (0)

LD = group with less diseases, MD = group with more diseases, GD = gastrointestinal diseases, IBS = Irritable Bowel Syndrome, BOS = Bacterial overgrowth syndrome, and UC = Ulcerative colitis.

**Table 4 foods-12-01233-t004:** Distribution of phenolic compounds intake by sex and age.

Participants (n)		ANOVA Main Effect of Group		
	PCI Comparisons	F/M, (M)	F (1, 962)	*p*
Sex F (798) M (175)	Fruits	122.32/133.54	8.65	0.003
	Vegetables	43.66/40.73	6.99	0.008
	Cereals	31.34/31.42	0.01	0.916
	Legumes	283.42/257.00	18.87	<0.001
	Seeds	193.61/176.99	12.32	<0.001
	Spices	62.04/64.77	2.74	0.10
	Beverages	724.30/684.77	6.91	0.009
	Mexican dishes	182.83/189.00	2.53	0.11
		(M)	F (4, 959)	*p*
Age 18–29 (436) 30–39 (262) 40–49 (156) 50–59 (81) >59 (29)	Fruits	18–29 (123.46) 30–39 (123.03) 40–49 (126.67) 50–59 (122.94) >59 (118.78)	0.34	0.85
	Vegetables	18–29 (41.57) 30–39 (42.94) 40–49 (45.17) 50–59 (44.79) >59 (44.14)	3.1	0.02
	Cereals	18–29 (32.02) 30–39 (31.16) 40–49 (30.90) 50–59 (30.60) >59 (27.56)	2.24	0.06
	Legumes	18–29 (288.88) 30–39 (275.56) 40–49 (270.44) 50–59 (259.95) >59 (251.97)	5.16	<0.001
	Seeds	18–29 (192.09) 30–39 (191.09) 40–49 (189.73) 50–59 (190.33) >59 (171.34)	0.93	0.44
	Spices	18–29 (59.14) 30–39 (63.42) 40–49 (65.20) 50–59 (70.37) >59 (68.97)	8.44	<0.001
	Beverages	18–29 (728.08) 30–39 (725.80) 40–49 (695.55) 50–59 (679.28) >59 (701.32)	2.10	0.08
	Mexican dishes	18–29 (184.88) 30–39 (183.22) 40–49 (180.36) 50–59 (184.18) >59 (194.28)	0.66	0.63

F = Female, M = Male, *n* = number of participants, and (M) = mean.

**Table 5 foods-12-01233-t005:** Regression analyses of diseases and PCI.

Disease	Variables	Coefficient Standardized			Model	ANOVA	
	Predictor	β	T	*p*-Value	R^2^	F	*p*-Value
LD/MD	Tomato	−0.2	−5.7	<0.001	0.5	78.5	<0.001
	Garlic	−0.2	−4.8	<0.001			
	Lettuce	−0.2	−4.7	<0.001			
	Corn	−0.1	−4.3	<0.001			
	Grape	−0.1	−3.1	0.002			
	Wine	−0.1	−2.4	0.02			
	Cranberry	0.1	2.4	0.01			
	Romeritos	−0.1	−3.1	0.002			
	Age	0.1	2.7	0.007			
	Wheat	0.1	2.9	0.002			
	Scholarship	−0.05	−2.3	0.02			
	Arroz con frijol	−0.05	−2.0	0.05			
Obesity	Tomato	−0.2	−4.5	<0.001	0.4	70.4	<0.001
	Corn	−0.2	−6.1	<0.001			
	Garlic	−0.2	−4.9	<0.001			
	Chamomile tea	−0.1	−2.4	0.01			
	Coffee	−0.1	−2.3	0.02			
	Grape	−0.1	−3.2	0.002			
	Plum	0.1	3.5	<0.001			
	Swiss chard	−0.1	−3.0	0.003			
	Enchiladas	−0.1	−2.3	0.02			
	Wine	−0.1	−2.3	0.02			
	Oregano	0.1	2.3	0.02			

PCI = Phenolic compounds intake; LD = less diseases and MD = more diseases.

## Data Availability

The data presented in this study are available on request from the corresponding author. The data are not publicly available, because they contain sensitive patient information.

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
