# Peer review of "Traditional Mexican Food: Phenolic Content and Public Health Relationship"

_foods, 2023, doi:10.3390/foods12061233_

Round 1
Reviewer 1 Report
The article is really very interesting and the link between phenolic compounds intake and health status is very important topic of food science and nutrition. Anyway the presented paper has very strong limitations, especially the design of the experimental part is not providing the precision and accuracy.
The main 2 strong limitations of the paper are.
1. Total phenolic compounds (TPC) of each food presented in Table 1 is not based on the own experiment, it is taken from the literature, different sources, different year of analysis which is making the paper week and unfortunately cannot provide the high level of precision and accuracy in order to have objective picture and scientific outcome.
2. Design of the survey is not accurate. Nine hundred seventy-three young adults (798 females; 165 males) between 18 and 79 years old were enrolled in this study. I do not believe that 79 years old is considered young in Mexico. Age sex group distribution is not justified, why such kind of heterogeneous distribution is available.
There also a lot of minor comments, but due to significant limitation of the study I do not consider that the paper can be published before the changes of the design of the study.
Author Response
We appreciate your comments.
Attached to the answers in the file.

Reviewer 2 Report
· Line no. 26-28, in abstract you have written the conclusion as “We conclude that the Mexican population should increase the consumption of foods associated with fewer diseases, particularly, Mexican dishes because they are good providers of phenolic compounds”. It seems to be more like a recommendation rather than a conclusion. Rewrite the conclusion appropriately.
· Line no. 31, it should be “has focused” instead of “has been focused”.
· Line no. 32, in introduction you are discussing about functional foods, phenolic compounds and health benefits. You can also validate this phenomenon of functionality of plant based foods due to phenolic compounds in combating diseases like diabetes and obesity from another article entitled as “Astounding Health Benefits of Jamun (Syzygium cumini) toward Metabolic Syndrome” (https://www.mdpi.com/1902832).
· Line no. 35, phenolic compounds are not micronutrients. These are plants based metabolites. Use the term “secondary metabolites” or “plant micronutrients”.
· Line no. 38, you are discussing about the anti-oxidant perspectives of plant based phenols. You can cite another paper entitled as “Delving into the Therapeutic Potential of Carica papaya Leaf against Thrombocytopenia” (https://www.mdpi.com/article/10.3390/molecules27092760).
· Line no. 39-48, you can also check the anti-inflammatory effects of flavonoids from the paper “Anti-inflammatory and anti-allergic potential of dietary flavonoids: A review” (https://www.sciencedirect.com/science/article/pii/S0753332222013348).
· Line no. 63-89, precise these paragraphs as there is repetition of diseases detail. In addition, introduction is becoming too long. Make it brief.
· Line no. 115-123, this paragraph is irrelevant to discuss here. Only describe the aim of your study at this point. The study description must be elaborated in section “Materials and Methods” while you can give recommendations in the “Conclusion” section.
· Line no. 126, you have used the term “young adults” but the age of your participants is between 18-79 years. Adults after 65 are categorized as older adults. You can use the term “adults” only.
· Line no. 126, don’t you think so that the selected age group (18-79) is too vast as with the progression of age, food choices, accessibility and digestibility vary greatly. So the results of survey must have been affected. Then you should also assess and validate these factors. You must categorize the results in special reference to younger and older adults.
· Line 127, you have selected 973 participants, out of which 798 are females and 165 are males. Then how will you justify the term “Mexican population” as your major percentage of subjects are females. In your survey there must be an equal or near to equal proportion of each gender to justify your study.
· Line no. 142, mention either to assess the frequency of taking PCs rich foods or not.
· Line no. 194, what do you mean by “within-subject factor”?
· Line no. 216, there is no equal distribution of sex in your study. Don’t you think so that your observation on sex basis has no base?
· In section 2.1, also mention that you have selected diseased participants (less or more diseased).
· Line no. 352, can you mention any reason of low intake of beverages and Mexican dishes in Mexican population?
· Line no. 461, did you observe any kind of association and incidence of CVDs with coffee intake?
· Line no. 526, conclusion is too short. Increase its length and also write your conclusion appropriately as concluding remarks not only as recommendations.
· Extensive grammatical and English language mistakes throughout the manuscript. Improve scientific writing.
Author Response

(The authors gave the same response as above.)

Reviewer 3 Report
1-The first thing that caught my attention in the article was that you gave the mushroom (Huitlacoche) as a vegetable in the table 1. It could be stated under the table in the form of an additional note that the mushroom is a plant (Fungi) but considered a vegetable.
2-The title to better reflect the content could have been:
‘Traditional Mexican Food-Phenolic content and Public Health Relationship’
3-Wouldn't it be more reliable if this important study was done primarily with clinical animals rather than humans? Maybe it can be made later and presented in Nutrients magazine.
4-The purpose of the work is good. Such studies should be in the literature. However, in the article, the names of the local dishes are given without being explained. After the mole is introduced, it will be much more satisfying if the question of what are the phenolic substances in the mole is answered.
For ex: Mole sauce, a sauce unique to Mexico, is often used in traditional Mexican dishes. Cocoa and spices come to the fore in this sauce, which is especially suitable for chicken meat. The ingredients are mixed until a smooth consistency is obtained. It can be stored by filling in can jars or bottles, if desired.
https://www.americastestkitchen.com/articles/1057-all-about-mole-what-it-is-where-it-comes-from-and-how-to-make-it
Author Response

(The authors gave the same response as above.)

Round 2
Reviewer 1 Report
The revised version doesn't provide sufficient justification to the raised issues highlighted during first revision. Anyway I do not consider with present research design and justification of aim and scope it is suitable for this journal.
Author Response
We attach a document with the answer.

Reviewer 2 Report
Well revised. Need proper formatting, which the journal manager and the team will do it.
Author Response
We appreciate your time and comments.